

# The identity of the South African toad *Sclerophrys capensis* Tschudi, 1838 (Amphibia, Anura)

Annemarie Ohler and Alain Dubois

Institut de Systématique, Évolution, Biodiversité, ISYEB—UMR7205—CNRS, MNHN, UPMC, EPHE, Muséum National d'Histoire Naturelle, Sorbonne Universités, Paris, France

## ABSTRACT

The toad species *Sclerophrys capensis* Tschudi, 1838 was erected for a single specimen from South Africa which has never been properly studied and allocated to a known species. A morphometrical and morphological analysis of this specimen and its comparison with 75 toad specimens referred to five South African toad species allowed to allocate this specimen to the species currently known as *Amietophrynus rangeri*. In consequence, the nomen *Sclerophrys* must replace *Amietophrynus* as the valid nomen of the genus, and *capensis* as the valid nomen of the species. This work stresses the usefulness of natural history collections for solving taxonomic and nomenclatural problems.

## INTRODUCTION

In zootaxonomy, early nomina (scientific names) often have a complex story. Many original type specimens have been lost and thus the status of the nomina cannot be assessed. In other cases, type specimens are present but have never received the necessary attention and been studied properly. This is the case of the amphibian nomen *Sclerophrys capensis* Tschudi, 1838, a toad from Cape of Good Hope described by *Tschudi (1838)* as a member of his family BOMBINATORES. This nomen is based on a single specimen (MNHN RA 0.742), whose holotype (holophoront) by original monotypy (monophory) has since then been kept in the collections of the Paris Muséum national d'Histoire naturelle (MNHN) where it has permanently been available for study by taxonomists.

The specimen was collected in the Cape Province of South Africa by Pierre Antoine Delalande and Jules Verreaux (then 12 years old). They made three field trips to the east and northwest of this province and collected 13,400 specimens, including 322 specimens of Reptiles and Amphibians (*Delalande, 1822*). The first travel to the east started on 11 November 1818 and was stopped by the battle of Grahamstown where 10,000 Xhosa people confronted English troops on 22 April 1819. A second trip started on 5 July 1819,

Corresponding author
Annemarie Ohler, ohler@mnhn.fr

led north to the Olifants River and enabled collection in the marshes of the Berg River. The third expedition started on 2 November 1819 and lasted 8 months. Delalande and his men went by boat to the Algoa Bay and then inland to the Keiskamma River. On 1 December 1820, Delalande returned back to France where he arrived at the end of the year (*Delalande, 1822*; *Barnard, 1956*; *Varley, 1956*).

Among the specimens collected by Delalande and Verreaux was a subadult toad which was described by *Tschudi (1838)* as a new genus and species *Sclerophrys capensis*. The nomen *Sclerophrys* was mentioned in the taxonomic review of volume 8 of the Erpétologie Générale (*Duméril & Bibron, 1841*, 300), but not in any synonymy list of amphibian species of this book. All *Bufo* specimens collected by Delalande were allocated by these authors to *Bufo pantherinus* Smith, 1828. However, apart from the holotype of *Sclerophrys capensis*, all the bufonid specimens collected by Delalande presently in the MNHN collection belong in the species now known as *Vandijkophrynus angusticeps* (Smith, 1848), but until 2006 known as *Bufo angusticeps*. The holotype of *Sclerophrys capensis* then was listed in the type catalogue of the Paris Museum of *Guibé* (*1950*, 15–16), who stated that by its characters it should be considered a member of the genus *Bufo* Garsault, 1764.

The nomina *Sclerophrys* and *S. capensis* are associated with a description and are therefore nomenclaturally available, not nomina nuda. They do not qualify either as nomina oblita, as they regularly appeared in the literature after the original description (*Duméril & Bibron, 1841*; *Knight, 1841*; *Owen, 1841*; *Agassiz, 1846*; *Knight, 1854*; *Cope, 1865*; *Saint-Lager, 1882*; *Saint-Lager, 1884*; *Neave, 1940*; *Guibé, 1950*; *Duellman & Trueb, 1985*; *Dubois & Bour, 2010*). However, they have not been considered in any recent taxonomic works on the Bufonidae (*Frost et al., 2006*; *Van Bocxlaer et al., 2009*; *Van Bocxlaer et al., 2010*; *Portik & Papenfuss, 2015*) or on the South African batrachofauna (*Du Preez & Carruthers, 2009*). *Dubois & Bour (2010)* mentioned the nomen *Sclerophrys* and stated that it applies to an African genus of bufonids, but postponed the publication of details to the completion of the present study.

The problem posed by the existence of a generic nomen of bufonids unallocated to a group of recent species was not a real one as long as the genus *Bufo* was not divided, but it becomes a real one if the genus has to be split, at least in subgenera. Among 33 synonyms of the generic nomen *Bufo* listed by *Duellman & Trueb* (*1985*, 533), *Sclerophrys* was the only one having an African type species, and it remained so until 2006: it was therefore bound to be the valid one for an African genus or subgenus of bufonid if such taxa had to be erected. We do not think that ignoring available nomina whenever they pose nomenclatural problems is good taxonomy: keeping such nomina as nomina dubia is providing the conditions for such nomina to become a problem whenever any taxonomist decides to unearth them. We consider that solving the *Sclerophrys capensis* nomenclatural problem should have been a preliminary action to take before coining new nomina for African bufonid genera that might possibly include this species. The same applies to the description of new species of this group or the resurrection of some specific nomina from synonymy, as has been the case in the recent decades. The aim of the present paper is to allocate unambiguously this specimen to a bufonid species present within the region visited by Delalande and Verreaux. For this, after unsuccessful attempts at obtaining

nucleic acids from this specimen, we used both a morphometrical and a morphological approach.

Based on the geographic origin of the specimen, there are two possibilities, which both would result in the invalidation of a nomen coined by *Frost et al. (2006)*: *Amietophrynus* if the nomen is finally applied to *Bufo pantherinus*, *Bufo pardalis* or *Bufo rangeri*, and *Vandijkophrynus* if it is applied to *Bufo angusticeps* or *Bufo gariepensis*. The nomina created in *Frost et al.*'s (*2006*) work are very recent and do not in the least qualify for conservation under the nomen protectum rule of the Code.

## MATERIAL AND METHODS

### Specimens studied

*Amietophrynus pantherinus* (Smith, 1828). Cape Flats: BMNH 1910.7.28.6, young. Cape of Good Hope: BMNH 1936.12.3.24, young, BMNH 1936.12.3.25, neotype of *Bufo pantherinus* Smith, 1828, adult female. Cape Peninsula: BMNH 1925.12.18.160–162, young. Hout Bay, near Cape Town: BMNH 1985.1438–1439, adult males. Near Cape Town: BMNH 1894.2.9.6, adult female. South Africa: BMNH 1846.6.18.28, young.

*Amietophrynus pardalis* (Hewitt, 1935). Cape Province: MNHN 4945, adult female; MNHN 5484, young female; MNHN 399, 3 juveniles. Port Elizabeth: BMNH 1871.4.21.10–11, adult females.

*Amietophrynus rangeri* (Hewitt, 1935). Cape Angulhas: ZFMK 85681, adult male. Johannesburg: ZFMK 85682, young male. Mossel Bay: ZFMK 85677, adult male; ZFMK 85672–6 and 85678–80, adult females. Port Elisabeth: NMW 5193, young. Tshipise, north Transvaal: NMW 26724, adult female.

*Sclerophrys capensis* Tschudi, 1838. Cape: MNHN 0.384, young male.

*Vandijkophrynus angusticeps* (Smith, 1848). Cape of Good Hope: BMNH 1858.11.25.160, lectotype of *Bufo angusticeps* Smith, 1848, adult female; MNHN 0.4941, adult male; MNHN 0.4947–4948, 1979.7807–7809 and 1994.1776–1778, adult females; NMW 5195.6–10, 5196.1–2 and 26728.5, adult females; NMW 5195.4–5, young males; NMW 5195.3, young. ZFMK 33168–9, adult males. Cape Angulhas: ZFMK 85735, adult male; ZFMK 85733, adult female. Fisantekraal: NMW 26728.4, adult male, NMW 26728.2, adult female; NMW 26728.3, young female. Longkloof: MNHN 0.384 and 1994.1779, adult females. Mossel Bay: ZFMK 85737, adult male. Strand-fontein: NMW 26728-1, adult female. Table Mountain: MNHN 1979.7806, adult male.

*Vandijkophrynus gariepensis* (Smith, 1848). Banks of the Orange (Gariep) River: BMNH 1858.11.25.157, lectotype of *Bufo gariepensis* Smith, 1848, young male. Augrabies: ZFMK 85711, adult male; ZFMK 85710 and 85712, adult females. Cape of Good Hope: NMW 5194.4, adult male; NMW 5197, adult female. Vioolsdrif: ZFMK 85702–5 and 85708, adult males; ZFMK 85701and 85706, adult females.

### Methods

A total of 75 specimens belonging to five species of South African bufonids, *Amietophrynus pantherinus* (2 males, 2 females, 4 young), *Amietophrynus pardalis* (3 females, 4 young), *Amietophrynus rangeri* (6 males, 5 females, 2 young) *Vandijkophrynus*

*angusticeps* (7 males, 23 females, 4 young) and *Vandijkophrynus gariepensis* (2 males, 9 females, 2 young), that occur in the Cape region, were studied and compared with the holotype of *Sclerophrys capensis*. On every specimen, 35 measurements (*Dubois & Ohler, 1999*) were taken by a single observer (AMO), either with a slide caliper or with an ocular micrometer (measurements smaller than 5 mm). In order to correct for size, every measurement was transposed into its logarithm and divided by the mean of the 35 logarithm-transposed measurements of the specimen (*Mosimann, 1970*). The ln-transformed variables were analysed using Discriminant Analysis. The holotype of *Sclerophrys capensis* was included without group membership for subsequent allocation to one of the groups.

The following morphological characters were used for morphological description of the specimens studied and for allocation of the holophoront of *Sclerophrys capensis* to a group (*Poynton & Lambiris, 1998*; *Du Preez & Carruthers, 2009*): spot on snout (distinct; indistinct; absent); spots on eyelids (paired spot; continuous band on head); skin on throat (granular; smooth); distal subarticular tubercle of finger III (a unique tubercle; paired tubercles); extension of web on outer side of toe III (number of phalanges free of web; see *Savage & Heyer, 1967*); fringes on digits (absent; narrow; broad); fringes on toes (absent; narrow; broad); separation of parotoid and eye (touching; separated); shape of parotoid (elongate; oval; broadened oval); parotoid height (flat; prominent); middorsal line (present; absent).

## RESULTS

### Description of the holotype of *Sclerophrys capensis Tschudi, 1838*

MNHN 0.742, young male; in rather bad state of conservation (Fig. 1). Specimen of medium size (SVL 41.2 mm), body rather robust.

Head of medium size, slightly larger (HW 17.0 mm) than long (HL 13.2 mm; MN 12.1 mm; MFE 9.6 mm; MBE 4.8 mm), flat above. Snout pointed, not protruding, its length (SL 5.4 mm) shorter than horizontal diameter of eye (EL 6.5 mm). Canthus rostralis rounded, loreal region slightly concave, acute. Interorbital space slightly concave, smaller (IUE 4.21 mm) than upper eyelid (UEW 4.54 mm) but larger than internarial distance (IN 2.72 mm); distance between front of eyes (IFE 7.1 mm) two thirds of distance between back of eyes (IBE 11.4 mm). Nostrils oval, closer to tip of snout (NS 2.59 mm) than to eye (EN 2.79 mm). Pupil indistinct. Tympanum (TYD 2.92 mm) oval, horizontal, indistinct, almost half of eye diameter, tympanum–eye distance (TYE 1.17 mm) almost half of tympanum diameter. Pineal ocellus absent. Vomerine ridge absent. Tongue shrunken.

Arm short, thin (FLL 10.1 mm), as long as hand (HAL 10.3 mm), not enlarged. Fingers thick (TFL 5.38 mm). Relative length, shortest to longest: II < IV < I < III. Tips of fingers rounded, without grooves. Fingers without dermal fringe; webbing absent. Subarticular tubercles very prominent, single, pointed, rounded. Prepollex oval; one prominent palmar tubercle; two supernumerary tubercles on base of each finger.

Shank four times longer (TL 17.2 mm) than its maximum width (TW 4.6 mm), shorter than thigh (FL 18.4 mm) and distance from base of internal metatarsal tubercle to tip

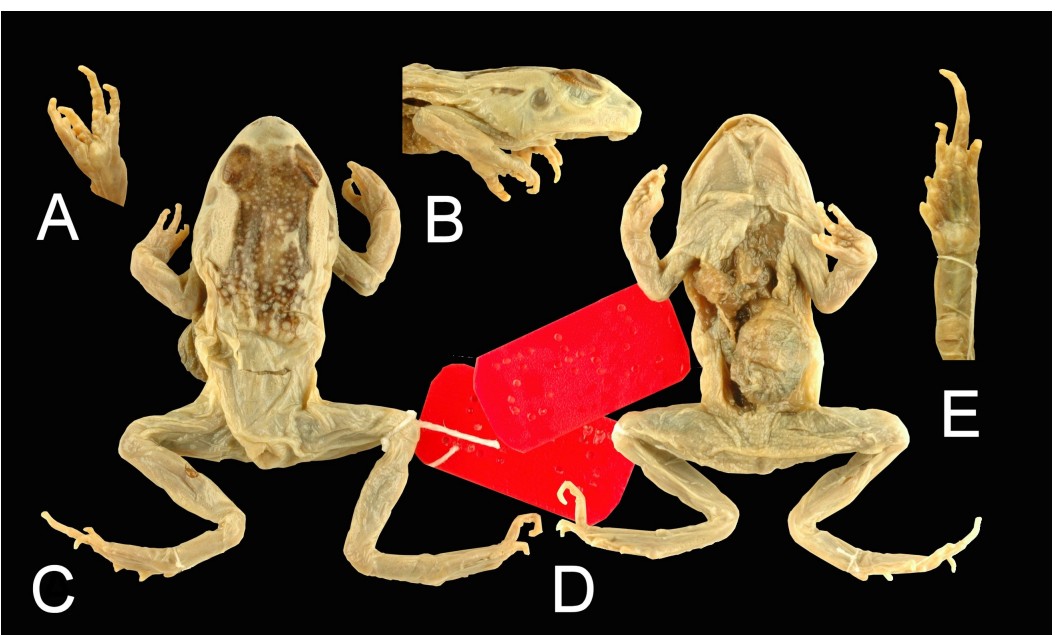

**Figure 1** *Sclerophrys capensis* **Tschudi, 1838.** MNHN 0.742, young male, holotype, SVL 41.2 mm. (A) ventral view of left hand; (B) lateral view of right side head and anterior body; (C) dorsal view of specimen; (D) ventral view of specimen; (E) ventral view of left foot.

of toe IV (FOL 18.7 mm). Toes rather long, rather thin; toe IV (FTL 10.7) one third of distance from base of tarsus to tip of toe IV (TFOL 27.8 mm). Relative length of toes, shortest to longest: I < II < V < III < IV. Tips of toes pointed, not enlarged, without grooves. Webbing formula: I 1–1 1/2 II 1–2 II 1 2/3–3 1/2 IV 3 1/2–2 V. (WTF 1.82; WFF 1.82; WI 1.17; WII 0.78; MTTF 8.7 mm; MTFF 9.1 mm; FTTF 9.6 mm; FFTF 10.5 mm). Dermal fringe along toe V absent. Subarticular tubercles prominent, rounded, all present. Inner metatarsal tubercle short, not prominent; its length (IMT 2.07 mm) 1.72 times in length of toe I (ITL 3.56 mm). Tarsal fold present, indistinct. Outer metatarsal tubercle present; several supernumerary tubercles on the sole of the foot.

Dorsal and lateral parts of head and body: snout smooth; between the eyes, back and flank with glandular warts. Large region, including upper eyelids and zone between the eyes and between the parotoids and anterior part of back of hardened aspect. Laterodorsal folds absent; supratympanic fold absent. Parotoid glands present, oval, elongate, perforate, more than two times longer (PL 10.4 mm) than wide (PW 4.1 mm), longer as distance between them (PD 8.6 mm), touching posterior border of eye. Cephalic ridges absent. (15) Co-ossified skin absent. Dorsal parts of limbs: forelimbs smooth; thigh, leg and tarsus with flattened glandular warts. Throat, chest, belly and thigh with glandular warts; foot smooth. Presence of macroglands: parotoid glands present.

Color of dorsal and lateral parts of head and body: whitish; "hardened region" dark brown. Dorsal parts of limbs: whitish. Ventral parts of head, body and limbs: whitish.

Male sexual characters. Presence of small testis. Nuptial spines and vocal sac openings absent.

**Table 1** **Results for Discriminant Function Analysis of 76 specimens of bufonid toads from South Africa including specimens allocated to *Amietophrynus pantherinus*, *A. pardalis*, *A. rangeri*, *Vandijkophrynus angusticeps*, *V. gariepensis* and *Sclerophrys capensis*, the latter included without group membership.** Eigenvalues of function 1–4 and their respective part in variance.

| | | Eigenvalues | | |
|---|---|---|---|---|
| Function | Eigenvalue | % of variance | Cumulative% | Canonical correlation |
| 1 | 17.060 | 70.5 | 70.5 | 0.972 |
| 2 | 4.339 | 17.9 | 88.4 | 0.901 |
| 3 | 1.882 | 7.8 | 96.1 | 0.808 |
| 4 | 0.933 | 3.9 | 100.0 | 0.695 |

**Table 2** **Results for Discriminant Function Analysis of 76 specimens of bufonid toads from South Africa including specimens allocated to *Amietophrynus pantherinus*, *A. pardalis*, *A. rangeri*, *Vandijkophrynus angusticeps*, *V. gariepensis* and *Sclerophrys capensis*, the latter included without group membership.** Significance of function 1–4 for discriminating groups.

| | | Wilks' lambda | | |
|---|---|---|---|---|
| Test of functions | Wilks' lambda | Chi-square | df | Sig. |
| 1 through 4 | 0.002 | 314.320 | 124 | 0.000 * |
| 2 through 4 | 0.034 | 169.634 | 90 | 0.000 * |
| 3 through 4 | 0.179 | 85.882 | 58 | 0.010 n.s. |
| 4 | 0.517 | 32.958 | 28 | 0.237 n.s. |

**Table 3** **Results for Discriminant Function Analysis of 76 specimens of bufonid toads from South Africa including specimens allocated to *Amietophrynus pantherinus*, *A. pardalis*, *A. rangeri*, *Vandijkophrynus angusticeps*, *V. gariepensis* and *Sclerophrys capensis*, the latter included without group membership.** Classification results giving predicted group memberships.

| Species | Predicted group membership | | | | | Total |
|---|---|---|---|---|---|---|
| | *angusticeps* | *gariepensis* | *rangeri* | *pardalis* | *pantherinus* | |
| *angusticeps* | 34 | 0 | 0 | 0 | 0 | 34 |
| *gariepensis* | 0 | 13 | 0 | 0 | 0 | 13 |
| *rangeri* | 0 | 0 | 13 | 0 | 0 | 13 |
| *pardalis* | 0 | 0 | 0 | 6 | 0 | 6 |
| *pantherinus* | 0 | 0 | 0 | 0 | 3 | 3 |
| *Sclerophrys* | 0 | 0 | 1 | 0 | 0 | 1 |

## Comparisons

In Discriminant Analysis, the percentage of variance accounted for was 70.5% on the first axis, 17.9% on the second axis (Table 1 and Fig. 2) and 7.8% on the third axis. The first axis had high loadings of the characters describing the body size, head shape, length of hand and leg, distance between parotoids and webbing, and proved useful for discrimination between the species (Chi$^2$ = 314.3; *df* = 124; *p* = 0.000) (Table 2). The second axis had high loadings of the characters describing distance between eyes and some characters measuring head shape, and also showed significant discrimination between the

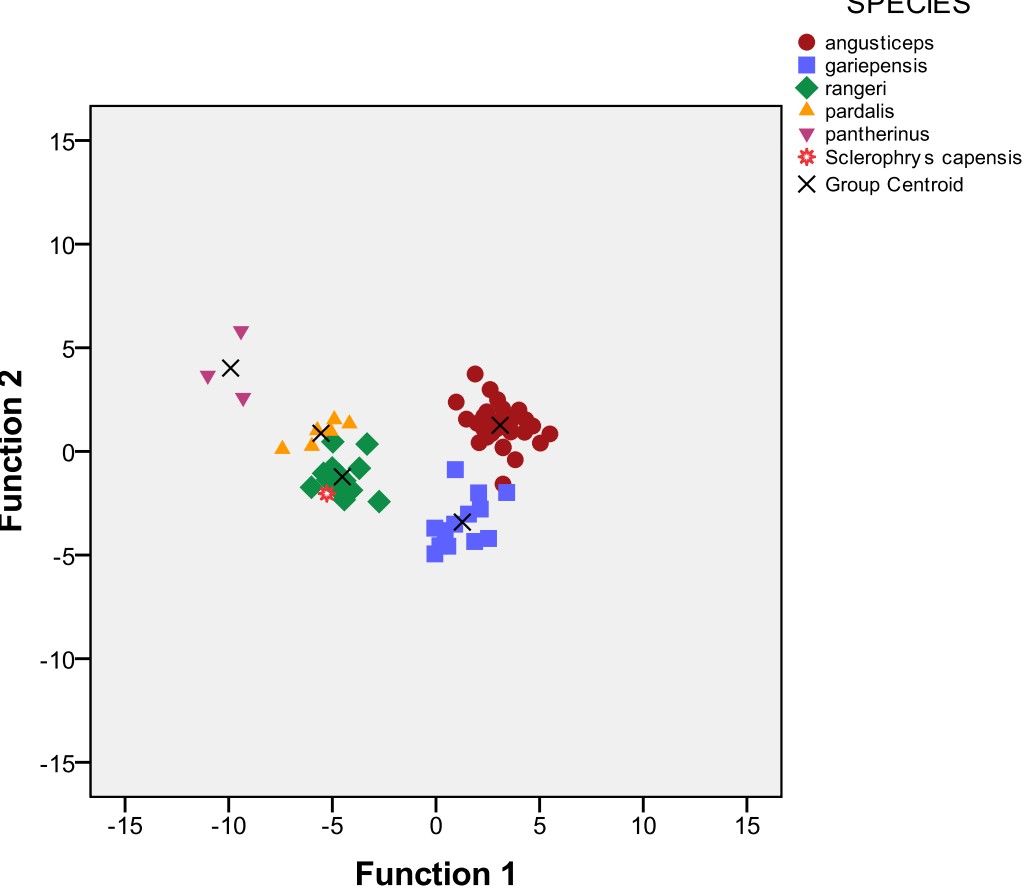

**Figure 2** Scatterplot of function 1 and 2 of Discriminant Function Analysis of 76 specimens of bufonid toads from South Africa including specimens allocated to *Amietophrynus pantherinus*, *A. pardalis*, *A. rangeri*, *Vandijkophrynus angusticeps*, *V. gariepensis* and *Sclerophrys capenis*, the latter included without group membership.

species (Chi$^2$ = 169.6; *df* = 90; *p* = 0.000). The third axis had high loadings concerning eye nostril distance, outer metatarsal tubercle length and webbing, but is not significantly discriminant between species (Chi$^2$ = 85.9; *df* = 58; *p* = 0.010). The analysis enabled separating *angusticeps*, *gariepensis* and *pantherinus* but there is some ambiguity in the distinction between *pardalis* and *rangeri*. The holotype of *Sclerophrys capensis*, included without group membership into the analysis, was clearly allocated to the group including *Amietophrynus rangeri* (Table 3 and Fig. 2).

The study of the morphological characters (Table 4) supports this allocation of the specimen of *Sclerophrys capensis*. This toad can be distinguished from *Vandijkophrynus angusticeps* by its granular skin on the throat (smooth with fine furrows in *V. angusticeps*), and by a single subarticular tubercle distally on finger III (this tubercle is double in *V. angusticeps*). The other bufonid species from the Cape region are morphologically very similar. *Vandijkophrynus gariepensis* shares a granular skin on throat and single

**Table 4** Morphological character states for bufonid species from South Africa allocated to *Amietophrynus pantherinus*, *A. pardalis*, *A. rangeri*, *Vandijkophrynus angusticeps*, *V. gariepensis* and *Sclerophrys capensis*.

| Characters | S. capensis | A. pantherinus | A. pardalis | A. rangeri | V. angusticeps | V. gariepensis |
|---|---|---|---|---|---|---|
| Spot on snout | – | Present | Present | Present, small | Present, sometimes indistinct | Indistinct or present |
| Spot on lids | Uniformly dark | Separate or continuous | Separate, continuous | Continuous | Separate | Separate, continuous |
| Skin on throat | Granular | Granular | Granular | Granular | Smooth | Granular |
| Distal SAT III | Single | Single | Single | Single | Double | Unique |
| Web toe III | 1 2/3 | 1 3/4–2 | 2 | 1 1/2–2 | 2–2 1/2 | 2–2 1/2 |
| Fringe on fingers | Absent | Present on finger II | Present on finger II | Present | Present | Present |
| Fringe on toes | Absent | Present | Present | Present | Present | Present |
| Parotoid eye | Touching | Touching | Touching | Usually separate | Separate, rarely touching | Separate, touching |
| Parotoid shape | Oval, elongate | Oval | Oval | Oval, elongate | Oval | Oval, elongate |
| Parotoid height | Rather prominent | Prominent | Prominent | Prominent | Prominent | Flat, prominent |
| Middorsal line | Absent | Present | Present | Absent | Present or absent | Absent |
| Dorsal pattern | Not visible | Large, paired spots, Perfectly distinct | Smaller paired spots, Poorly distinct | Small spots, Distinct | Small irregular spots, Indistinct | Small irregular spots, Distinct or indistinct |

subarticular tubercles with the *Amietophrynus* species. The other characters are poorly discriminant among the species. *A. pantherinus* can be distinguished by its very distinct dorsal pattern consisting of large paired neatly outlined spots. These spots are paired but smaller and not neatly outlined in *A. pardalis*. In *A. rangeri*, the pair of spots of the upper eyelids forms an uninterrupted band. Such a band can be present in the other species but in them this character state is much rarer. A middorsal line is absent in *V. gariepensis* and *A. rangeri* but present in *A. pantherinus* and *A. pardalis*. *Sclerophrys capensis* is clearly an *Amietophrynus* as it has granular skin on the throat and single subarticular tubercles on fingers and toes. Its allocation to a species is tenuous on morphological evidence because the most discriminating characters are coloration and colour pattern and these are faded in this old specimen. The dark spot in the head region of the *Sclerophrys* specimen does not show any lighter band in its middle so it might indicate that there was no separation of the lid spots and no middorsal line. Less than 2 phalanges are free of webbing on the outer side of toe III. These character states are concordant with *Sclerophrys capensis* being an *Amietophrynus rangeri*.

## Taxonomic conclusion

The holotype of *Sclerophrys capensis* is allocated to *Bufo rangeri* by morphometric analysis. It shares character states with this species and other species presently allocated

to *Amietophrynus*. Thus *Sclerophrys* is a subjective senior synonym of *Amietophrynus* and is the valid nomen of the genus. This nomen has been used after 1899 and its junior synonym *Amietophrynus* is a very recent nomen, published in 2006, and thus does not comply with the requirements of the Code for reversal of reference. *Bufo regularis rangeri* was described as a subspecies by *Hewitt (1935)* and has been considered since *Poynton (1964)* as a valid taxon at the species level. The validity of the species was never discussed, and this nomen appears in all recent lists of amphibians of South Africa. Searching on the 'Web of Science,' we found 29 references (including only four in which the nomen appears in the title of scientific works). However, the specific nomen *capensis* has been used four times in the literature after 1899 (see above). Therefore the conditions are not met to consider *rangeri* as a nomen protectum, and this nomen should be replaced by *capensis*. Thus the valid nomen of the toad currently known as *Amietophrynus rangeri* (Hewitt, 1935) is *Sclerophrys capensis* Tschudi, 1838.

The following nomenclatural changes are a consequence of the synonymy of *Sclerophrys capensis* with *Amietophrynus rangeri*:

### *Sclerophrys* Tschudi, 1838

Type-species by original monotypy (monophory): *Sclerophrys capensis* Tschudi, 1838. Grammatical gender: feminine.

### Synonym:

*Amietophrynus* Frost et al., 2006. Type-species by original designation: *Bufo regularis* Reuss, 1833. Grammatical gender: masculine.

### Included species:

*Sclerophrys arabica* (Heyden, 1827); *Sclerophrys asmarae* (Tandy, Bogart, Largen & Feener, 1982); *Sclerophrys blanfordii* (Boulenger, 1882); *Sclerophrys brauni* (Nieden, 1911); *Sclerophrys buchneri* (Peters, 1882); *Sclerophrys camerunensis* (Parker, 1936); *Sclerophrys capensis* Tschudi, 1838; *Sclerophrys channingi* Barej, Schmitz, Menegon, Hillers, Hinkel, Böhme & Rödel, 2011; *Sclerophrys chudeaui* (Chabanaud, 1919); *Sclerophrys cristiglans* (Inger & Menzies, 1961); *Sclerophrys danielae* (Perret, 1977); *Sclerophrys djohongensis* (Hulselmans, 1977); *Sclerophrys dodsoni* (Boulenger, 1895); *Sclerophrys fuliginata* (de Witte, 1932); *Sclerophrys funerea* (Bocage, 1866); *Sclerophrys garmani* (Meek, 1897); *Sclerophrys gracilipes* (Boulenger, 1899); *Sclerophrys gutturalis* (Power, 1927); *Sclerophrys kassasii* (Baha El Din, 1993); *Sclerophrys kerinyagae* (Keith, 1968); *Sclerophrys kisoloensis* (Loveridge, 1932); *Sclerophrys langanoensis* (Largen, Tandy & Tandy, 1978); *Sclerophrys latifrons* (Boulenger, 1900); *Sclerophrys lemairii* (Boulenger, 1901); *Sclerophrys maculata* (Hallowell, 1854); *Sclerophrys mauritanica* (Schlegel, 1841); *Sclerophrys pantherina* (Smith, 1828); *Sclerophrys pardalis* (Hewitt, 1935); *Sclerophrys pentoni* (Anderson, 1893); *Sclerophrys perreti* (Schiøtz, 1963); *Sclerophrys poweri* (Hewitt, 1935); *Sclerophrys reesi* (Poynton, 1977); *Sclerophrys regularis* (Reuss, 1833); *Sclerophrys steindachneri* (Pfeffer, 1893); *Sclerophrys superciliaris* (Boulenger, 1888); *Sclerophrys taiensis* (Rödel & Ernst, 2000); *Sclerophrys tihamica* (Balletto & Cherchi, 1973); *Sclerophrys togoensis* (Ahl, 1924); *Sclerophrys tuberosa* (Günther, 1858); *Sclerophrys turkanae* (Tandy & Feener, 1985);

*Sclerophrys urunguensis* (Loveridge, 1932); *Sclerophrys villiersi* (Angel, 1940); *Sclerophrys vittata* (Boulenger, 1906); *Sclerophrys xeros* (Tandy, Tandy, Keith & Duff-MacKay, 1976).

This work stresses the importance of natural history collections for resolving taxonomic and nomenclatural problems. As shown here this can be done using morphological information only and do not always require the recourse to molecular data: compare *Cappellini et al. (2014)* and *Dubois, Nemésio & Bour (2014)*.

**Abbreviations for Museum Collections**

| | |
|---|---|
| **BMNH** | Natural History Museum, London, United Kingdom |
| **MNHN** | Muséum National d'Histoire Naturelle, Paris, France |
| **NMW** | Naturhistorisches Museum, Wien, Austria |
| **ZFMK** | Zoologisches Forschungsinstitut und Museum Alexander Koenig, Bonn, Germany |

## ACKNOWLEDGEMENTS

For the loan and access to specimens we thank Jeff Streicher (BMNH), Laure Pierre and staff (MNHN), Heinz Grillitsch and Georg Gaßner (NMW), and Claudia Koch and Wolfgang Böhme (ZFMK). Nalani Schnell (MNHN) helped for the preparation of the figure of the holotype. We acknowledge the reviewers for careful reading and fruitful suggestions.

### Funding

This work was supported by funding from UMR 7205 ISYEB CNRS, MNHN, UPMC, EPHE, and Sorbonne Universités. The funders had no role in study design, data collection and analysis, decision to publish, or preparation of the manuscript.

### Grant Disclosures

The following grant information was disclosed by the authors:
UMR 7205 ISYEB CNRS.
MNHN.
UPMC.
EPHE.
Sorbonne Universités.

### Competing Interests

Alain Dubois is an Academic Editor for PeerJ.

### Author Contributions

- Annemarie Ohler conceived and designed the experiments, performed the experiments, analyzed the data, contributed reagents/materials/analysis tools, wrote the paper, prepared figures and/or tables.
- Alain Dubois conceived and designed the experiments, wrote the paper, reviewed drafts of the paper.

## Animal Ethics

The following information was supplied relating to ethical approvals (i.e., approving body and any reference numbers):

As it concerns historical animals, no experimentation with live animals was done.

## Data Availability

Raw data is available in the Supplemental Information.

## Supplemental Information

Supplemental information for this article can be found online at http://dx.doi.org/10.7717/peerj.1553#supplemental-information.

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
