# Peer review of "The identity of the South African toad Sclerophrys capensis Tschudi, 1838 (Amphibia, Anura)"

_PeerJ, doi:10.7717/peerj.1553_

## Round 0.1 · original submission · Minor Revisions

· Academic Editor

Minor Revisions

The comments by both reviewers are very positive and the changes they recommend are minor and should be very easy to implement. In fact, I've never personally handled such a clean manuscript before - congratulations. Once you make the changes recommended by the reviewers, I will be happy to accept the manuscript for publication in PeerJ.

·

Basic reporting

Carefully conducted study.

Experimental design

Appropriately conducted multivariate morphometrical procedure and analysis.

Validity of the findings

Valid.

Additional comments

This is a very direct and well-done manuscript that conclusively shows that a long-ignored type specimen can be allocated to a species currently known by a different name. The biological entity has a small literature and the change in name is likely to cause no substantive difficulty. A bit more troublesome is that fact that the generic nomen is the oldest for any toad endemic to Africa, and thus authors replace a name, Amietophrynus, applied less than 10 years ago to a relatively large clade of African toads. The authors of Amietophrynus failed to exercise due diligence and thus did not examine the existing type specimen of Sclerophrys capensis in the Paris museum. This inattention is the direct cause of the necessity of the name changes made by the authors of the manuscript under consideration. There is no need to consider appeals for conservation of current taxonomy.

In the interests of improving the English composition of the generally well-written manuscript, I offer the following suggestions:

Abstract
Line 4. Change from “allowed to allocate”, which is awkward English, to “allowed allocation of”

Introduction
Line 15. Cape rather than Cap
Line 19. “it” rather than “is”
Line 24. Replace “The first travel –“ with “The first trip – “
Line 25. Rewrite as follows from start of line: in which 20,000 Xhosa people confronted English troops on 22 April 1819. A second trip started on
Line 26. Replace “allowed” with “enabled”
Line 37. It would be appropriate to indicate that at the time Guibé wrote, Vandijkophrynus angusticeps also was placed in Bufo.
Line 59. Replace “et” with “and”
Line 60. Replace “a both” with “both a”
Line 98. Drop word “Thus”, which has only an inferred antecedent, and start the sentence with “A”
Line 128. Replace “shrinked” with “shrunken” (“shrinked” is exceedingly rare as an English word – used by novelists: “She shrinked from violence”).
Line 132. “surnumerary” I doubt this is a word in English or Latin. Not in my dictionaries. Is supernumerary meant?
Line 161. Has or had? Both work, but for the 1st and 2nd axis, had is used, so be consistent.
Line 163. Replace “allowed” with “enabled”

Reviewer 2 ·

Basic reporting

Well written and flows well. Takes the reader through the logic of the authors as well as on the 'adventure' of the early expedition.

Experimental design

"No Comments"

Validity of the findings

The findings are significant in that it merits the taxonomic change of the recent Amietophrynus to the more senior Sclerophys based on this specimen.

Additional comments

To Author
The manuscript on the taxonomic validy of Sclerophrys capensis and the genus Sclerophys vs Amietophrynus are discussed by Ohler and Dubois. This paper is a fantastic example of the utility of natural history collections, excellent field documentation by collectors and challenging the identification of species that are discussed in the literature. I enjoyed reading this manuscript, it is not only clear in its taxonomic implications, but reads very nicely like a detective story. The analyses performed by the authors firmly place S. capensis as A. rangeri and am in accord with their conclusion of replacing Amietophrynus by Sclerophys based on the latter being an earlier used nomen. My comments are relatively minor.
- Abstract: morphometrical → morphometric
- Abstract: “…toad species allowed to allocate…” should read “…toad species, allowed us to ….”
- Line 15: Cap → Cape
- Lines 17-20: combine sentences and edits, should read as follows to simplify and increase flow: “…specimen (MNHN RA 0.742) whose holotype (holophoront), by original monotypy (monophory), has been kept in the collections of the Paris Muséum national d’Histoire naturelle (MNHN) where it has been available for study by taxonomists.”
- Line 24-25, edits: “…Grahamstown where 10,000 Xhosa people confronted English…”
- Line 26: 1919 should be 1819
- Line 32: Bufo should be in italics
- Line 62: geographical should be geographic
- Line 101: shold read “…in the Cape region…”
- Line 103: should be spelled ‘caliper’
- Line 114: hight → height
- Figure 1: This is purely stylistic, but the letters in the figure are in a very odd order, the figure may flow better if CD were below A and B… which would also lead to the authors having to modify literature citation of this figure
- Line 124: should be ‘internarial’
- Line 132: should be ‘supernumerary’
- Line 134: “legs four times longer…than wide” this is a very strangely phrased measurement, which aspect is being measured… pelvis to toe tip? Femur only? What is being referred to as height? Dorsal to ventral width of leg along its mid length? The legs generally taper. More details should be provided here.
- Line 138: literature citation for use of webbing formula?
- Line 170: where is the closing parenthesis for “…(this tubercle…)…”
- Line 179: delicate correct word?
- Line 186: morphometrical → morphometric
- Line 187: characters → character
- Line 195: capensis should bein italics
-

---

## Round 0.2 · accepted · Accept

· Academic Editor

Accept

Thank you for your diligence in addressing the reviewers' suggestions. I am happy to accept your manuscript for publication in PeerJ.

Although it is of course up to you whether to publish the review history alongside your paper, I think that doing so would add value to the paper and to science in general. It would be nice to point to in future discussions about the utility of peer review - and in teaching students - as an example of a manuscript that was already scientifically solid getting further tightened up by going through the peer-review process.